# Latent Dynamic Factor Analysis of High-Dimensional Neural Recordings

**Heejong Bong**,* **Zongge Liu**\*
Carnegie Mellon University
hbong, zonggel@andrew.cmu.edu

**Zhao Ren**
University of Pittsburgh
zren@pitt.edu

**Matthew A. Smith, Valérie Ventura, Robert E. Kass**
Carnegie Mellon University
msmith, vventura, kass@andrew.cmu.edu

## Abstract

High-dimensional neural recordings across multiple brain regions can be used to establish functional connectivity with good spatial and temporal resolution. We designed and implemented a novel method, Latent Dynamic Factor Analysis of High-dimensional time series (LDFA-H), which combines (a) a new approach to estimating the covariance structure among high-dimensional time series (for the observed variables) and (b) a new extension of probabilistic CCA to dynamic time series (for the latent variables). Our interest is in the cross-correlations among the latent variables which, in neural recordings, may capture the flow of information from one brain region to another. Simulations show that LDFA-H outperforms existing methods in the sense that it captures target factors even when within-region correlation due to noise dominates cross-region correlation. We applied our method to local field potential (LFP) recordings from 192 electrodes in Prefrontal Cortex (PFC) and visual area V4 during a memory-guided saccade task. The results capture time-varying lead-lag dependencies between PFC and V4, and display the associated spatial distribution of the signals.

## 1 Introduction

New electrode arrays for recording electrical activity generated by large networks of neurons have created great opportunities, but also great challenges for statistical machine learning (e.g., Steinmetz et al., 2018). For example, Local Field Potentials (LFPs) are signals that represent the bulk activity in relatively small volumes of tissue (Buzsáki et al., 2012; Einevoll et al., 2013), and they have been shown to correlate substantially with the BOLD fMRI brain imaging signal (Logothetis et al., 2001; Magri et al., 2012). Typical LFP data sets may have dozens to hundreds of time series in each of two or more brain regions, recorded simultaneously across many experimental trials. A motivating example in this paper is LFP recordings from a prefrontal cortex (PFC) and visual area V4 during a visual working memory task. V4 has been reported to retain higher-order information (e.g., color and shape) and attention in visual processing (Fries et al., 2001; Orban, 2008), while PFC is considered to exert cognitive control in working memory (Miller and Cohen, 2001). Despite their spatial distance and functional difference, these regions have been presumed to cooperate during visual working memory tasks. Various approaches have been used to track the interaction among brain regions Adhikari et al. (2010); Buesing et al. (2014); Gallagher et al. (2017); Hultman et al. (2018); Jiang et al. (2015). In particular, delay-specific theta synchrony led by PFC has been discovered during visual memory tasks (Liebe et al., 2012; Sarnthein et al., 1998).

---

Because many functional interactions among brain regions are transient, it is highly desirable to have methods that accommodate non-stationary behavior in the multivariate time series recorded in each region. We report here an extension of Gaussian process factor analysis (GPFA, Yu et al., 2009) to two or more groups of time series, where the main interest is non-stationary cross-group interaction; furthermore, the multivariate noise within groups can have both spatial covariation and non-stationary temporal covariation. Here, spatial covariation refers to dependence among the time series and, in the neural context, this results from the spatial arrangement of the electrodes, each of which records one of the time series. Our approach uses probabilistic CCA, but the framework allows rich spatiotemporal dependencies. These generalizations come at a cost: we now have a high-dimensional time series problem within each brain region together with a high-dimensional covariance structure. We solve these high-dimensional problems by imposing sparsity of the dominant effects, building on Bong et al. (2020), which treats the high-dimensional covariance structure in the context of observational white noise, and by incorporating banded covariance structure as in Bickel and Levina (2008). We thus call our method Latent Dynamic Factor Analysis of High-dimensional time series, LDFA-H.

In a simulation study, based on realistic synthetic time series, we verify the recovery of cross-region structure even when some of our assumptions are violated, and even in the presence of high noise. We then apply the method to 192 LFP time series recorded simultaneously from both Prefrontal Cortex (PFC) and visual area V4, during a memory task, and find time-varying cross-region dependencies.

## 2   Latent Dynamic Factor Analysis of High-dimensional time series

We treat the case of two groups of time series observed, repeatedly, $N$ times. Let $X^1_{:,t} \in \mathbb{R}^{p_1}$ and $X^2_{:,t} \in \mathbb{R}^{p_2}$ be $p_1$ and $p_2$ recordings at time $t$ in each of the two groups, for $t = 1, \ldots, T$. As in Yu et al. (2009), we assume that a $q$-dimensional latent factor $Z^k_{:,t} \in \mathbb{R}^q$ drives each group, here, each brain region, according to the linear relationship

$$X^k_{:,t} \mid Z^k_{:,t} = \mu^k_{:,t} + \beta^k \cdot Z^k_{:,t} + \epsilon^k_{:,t}, \tag{1}$$

for brain region $k = 1, 2$, where $\mu^k_{:,t} \in \mathbb{R}^{p_k}$ are mean vectors, $\beta^k \in \mathbb{R}^{p_k \times q}$ are matrices of constant factor loadings, and $\epsilon^k_{:,t} \in \mathbb{R}^{p_k}$ are errors centered at zero (independently of the latent vectors $Z$). We are interested in the pairwise cross-group dependencies of the latent vectors $Z^1_{f,:}$ and $Z^2_{f,:}$, for $f = 1, \ldots, q$. As in (Bong et al., 2020), we assume that the time series of these latent vectors follows a multivariate normal distribution

$$\begin{pmatrix} Z^1_{f,:} \\ Z^2_{f,:} \end{pmatrix} \sim \mathrm{MVN}(0, \Sigma_f), \quad f = 1, \ldots, q, \tag{2}$$

where $\Sigma_f$ describes all of their simultaneous and lagged dependencies, both within and between the two vectors. We assume the $N$ sets of random vectors $(\epsilon, Z)$ are independent and identically distributed. Fig. 1a illustrates the dependence structure of this model. We let $\mathrm{P}_f$ be the correlation matrix corresponding to $\Sigma_f$, and write its inverse as

$$\Pi_f = \mathrm{P}_f^{-1} = \left( \begin{array}{c|c} \Pi^{11}_f & \Pi^{12}_f \\ \hline \Pi^{12\top}_f & \Pi^{22}_f \end{array} \right) \tag{3}$$

where $\Pi^{11}_f$ and $\Pi^{22}_f$ are the scaled auto-precision matrices and $\Pi^{12}_f$ is the scaled cross-precision matrix. We now assume finite-range partial auto-correlation and cross-correlation for $(Z^1_{f,t}, Z^2_{f,t})$, so that $\Pi^{11}_f$, $\Pi^{22}_f$ and $\Pi^{12}_f$ in Equation (3) have a banded structure. Specifically, for $k, l = 1, 2$, we assume there is a value $h^{kl}_f$ such that $\Pi^{kl}_f$ is a $(2h^{kl}_f + 1)$-diagonal matrix. Because our goal is to address the cross-region connectivity and lead-lag relationship, we are particularly interested in the estimation of $\Pi^{12}_f$ for each latent factor $f = 1, \ldots, q$. Note that the non-zero elements $\Pi^{12}_{f,(t,s)}$, depicted as the red star in the expanded display within Fig. 1b, determine associations between the latent pair $Z^1_{f,:}$ and $Z^2_{f,:}$, which are simultaneous when $t = s$ and lagged when $t \neq s$.

Finally, we model the noise in Eq. (1) as a Gaussian random vector

$$\mathrm{Vec}(\epsilon^k) = (\epsilon^k_{:,1}; \epsilon^k_{:,2}; \ldots; \epsilon^k_{:,T}) \sim \mathrm{MVN}(0, \Phi^k), \quad k = 1, 2, \tag{4}$$

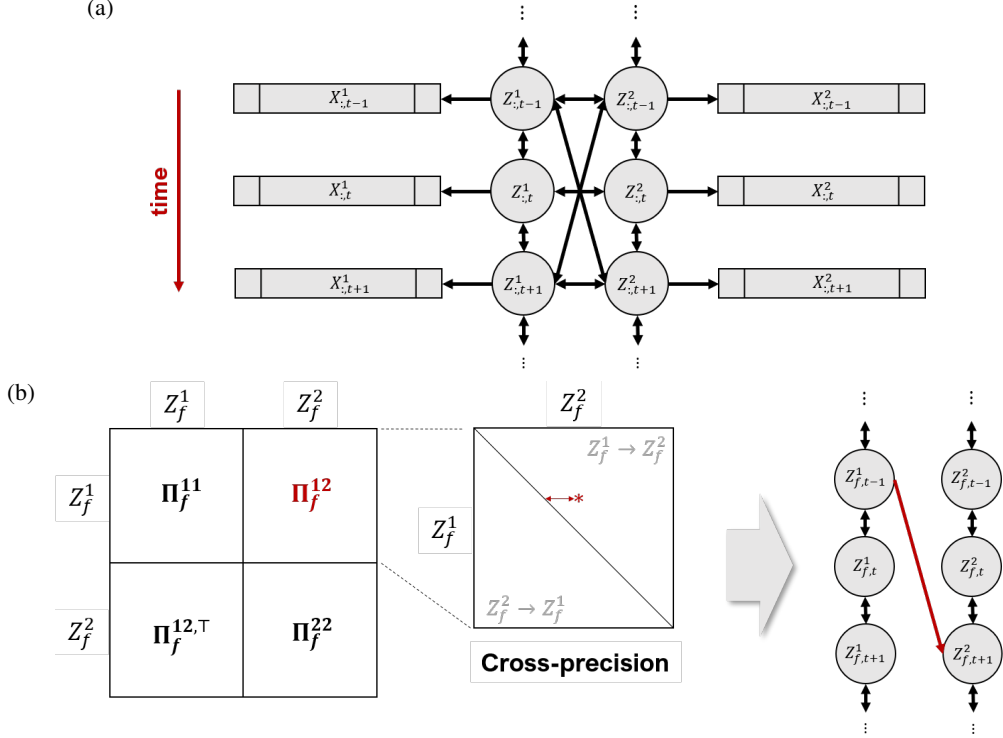

Figure 1: **LDFA-H model. (a)** *Dynamic associations between vectors $X^1_{:,t}$ and $X^2_{:,s}$ are summarized by the dynamic associations between their associated 1D latent variables $Z^1_{:,t}$ and $Z^2_{:,s}$.* **(b)** *When a significant cross-precision entry is identified, e.g., the red star in the expanded view of $\Pi^{12}_f$, its coordinates and distance from the diagonal indicate at what time in the experiment connectivity between two brain areas occurs, and at what lead or lag. Here the red star is in the upper diagonal of $\Pi^{12}_f$, which means that, at this particular time, region 1 leads region 2, or $Z^1_f \rightarrow Z^2_f$ in short (a non-zero entry in the lower diagonal would mean $Z^2_f \rightarrow Z^1_f$). We represent this association by the red arrow on the right-most plot, with a lag of two units of time for illustration.*

where we allow $\Phi^k$ to have non-zero off-diagonal elements to account for within-group spatiotemporal dependence. We assume $\Phi^k$ can be written in Kronecker product form

$$\Phi^k = \Phi^k_{\mathcal{T}} \otimes \Phi^k_{\mathcal{S}}, \; k = 1, 2, \tag{5}$$

where $\Phi^k_{\mathcal{T}}$ and $\Phi^k_{\mathcal{S}}$ are the temporal and spatial components of $\Phi^k$, as is often assumed for spatiotemporal matrix-normal distributions, e.g., (Dawid, 1981). Although this is a strong assumption, implying, for instance, that the auto-correlation of every $X^k_{i,:}$ is proportional to $\Phi^k_{\mathcal{T}}$, we regard $\Phi_k$ as a nuisance parameter: our primary interest is $\Sigma_f$ in Eq. (2). We also assume an auto-regressive order at most $h^k_\epsilon$, so that $\Gamma^k_{\mathcal{T}} = \left(\Phi^k_{\mathcal{T}}\right)^{-1}$ is a $(2h^k_\epsilon + 1)$-diagonal matrix. In our simulation we show that we can recover $\Sigma_f$ accurately even when the Kronecker product and bandedness assumptions fail to hold.

The model in Equations (1)-(5) generalizes other known models. First, when $q = 1$, and $Z^1 = Z^2$ remains constant over time, in the noiseless case ($\epsilon_k = 0$), it reduces to the probabilistic CCA model of Bach and Jordan (2005); see Theorem 2.2 of Bong et al. (2020) Thus, model (1)-(5) can be viewed as a denoising, multi-level and dynamic version of probabilistic CCA. Second, when $k = 1$, the Gaussian processes are stationary, and the $\epsilon$ vectors are white noise, (1)-(5) reduces to GPFA (Yu et al., 2009). Thus, (1)-(5) is a two-group, nonstationary extension of GPFA that allows for within-group spatio-temporal dependence.

**Identifiability and sparsity constraints** Despite the structure imposed on $\Phi_k$ in Eq. (5), parameter identifiability issues remain. Our model in Eqs. (1), (2) and (4) induces the marginal distribution of

the observed data $(X^1, X^2)$:

$$\left(X_{:,1}^1; X_{:,2}^1; \ldots; X_{:,T}^2\right) \sim \mathrm{N}\left((\mu_{:,1}^1; \mu_{:,2}^1; \ldots; \mu_{:,T}^2), S\right) \tag{6}$$

where $S$ is the marginal covariance matrix given by:

$$S = \begin{bmatrix} \Phi_{\mathcal{T}}^1 \otimes \Phi_{\mathcal{S}}^1 & 0 \\ 0 & \Phi_{\mathcal{T}}^2 \otimes \Phi_{\mathcal{S}}^2 \end{bmatrix} + \sum_{f=1}^{q} \begin{bmatrix} \Sigma_f^{11} \otimes (\beta_f^1 \beta_f^{1\top}) & \Sigma_f^{12} \otimes (\beta_f^1 \beta_f^{2\top}) \\ \Sigma_f^{12\top} \otimes (\beta_f^2 \beta_f^{1\top}) & \Sigma_f^{22} \otimes (\beta_f^2 \beta_f^{2\top}) \end{bmatrix}. \tag{7}$$

The family of parameters

$$\theta^{\{\alpha^1, \alpha^2\}} = \left\{ \begin{array}{l} \Sigma_1^{\{\alpha_1^1, \alpha_1^2\}}, \ldots, \Sigma_q^{\{\alpha_q^1, \alpha_q^2\}},\ \Phi_{\mathcal{S}}^1 - \sum_{f=1}^{q} \alpha_f^1 \beta_f^1 \beta_f^{1\top},\ \Phi_{\mathcal{S}}^2 - \sum_{f=1}^{q} \alpha_f^2 \beta_f^2 \beta_f^{2\top}, \\[2mm] \Phi_{\mathcal{T}}^1,\ \Phi_{\mathcal{T}}^2,\ \beta^1,\ \beta^2,\ \mu^1,\ \mu^2 \end{array} \right\}, \tag{8}$$

where $\Sigma_f^{\{\alpha_f^1, \alpha_f^2\}} = \left\{ \Sigma_f + \begin{bmatrix} \alpha_f^1 \Phi_{\mathcal{T}}^1 & 0 \\ 0 & \alpha_f^2 \Phi_{\mathcal{T}}^2 \end{bmatrix} \right\}$, induce the same marginal distribution in Eq. (6),
for all $\alpha^1, \alpha^2 \in \mathbb{R}^q$ (notice that $\theta = \theta^{\{0,0\}} = \{\Sigma_1, \ldots, \Sigma_q, \Phi_{\mathcal{S}}^1, \Phi_{\mathcal{S}}^2, \Phi_{\mathcal{T}}^1, \Phi_{\mathcal{T}}^2, \beta^1, \beta^2, \mu^1, \mu^2\}$
is the original parameter). Preliminary analysis of LFP data indicated that strong cross-region
dependence occurs relatively rarely. We therefore resolve this lack of identifiability by choosing the
solution given by maximizing the likelihood with an L1 penalty, under the assumption that the inverse
cross-correlation matrix $\Pi_f^{12}$ is a sparse $(2h_f^{12} + 1)$-diagonal matrix.

**Latent Dynamic Factor Analysis of High-dimensional time series (LDFA-H)**   Given $N$ simul-
taneously recorded pairs of neural time series $\{X^1[n], X^2[n]\}_{n=1,\ldots,N}$, the maximum penalized
likelihood estimator (MPLE) of the inverse correlation matrix of the latent variables solves

$$\left(\widehat{\Pi}_1, \ldots, \widehat{\Pi}_q\right) = \mathrm{argmin} \ -\frac{1}{N} \sum_{n=1}^{N} l\left(\theta; X^1[n], X^2[n]\right) + \sum_{f=1}^{q} \sum_{k,l=1}^{2} \left\| \Lambda_f^{kl} \odot \Pi_f^{kl} \right\|_1 \tag{9}$$

$$\text{s.t. } \Gamma_{\mathcal{T}}^k \text{ is } (2h_\epsilon^k + 1)\text{-diagonal,}$$

where the log-likelihood is

$$l\left(\theta; X^1, X^2\right) = -\log \det S - (X_{:,1}^1 - \mu_{:,1}^1; \ldots; X_{:,T}^2 - \mu_{:,T}^2)^\top S^{-1} (X_{:,1}^1 - \mu_{:,1}^1; \ldots; X_{:,T}^2 - \mu_{:,T}^2), \tag{10}$$

with $S$ defined in Eq. (7), and the constraints are

$$\Lambda_{f,(t,s)}^{kl} = \begin{cases} \infty, & (t,s) : |t-s| > h_f^{kl}, \\ \lambda_f, & (t,s) : 0 < |t-s| \le h_f^{kl}, \ k \ne l, \\ 0, & \text{otherwise.} \end{cases} \tag{11}$$

for factor $f = 1, \ldots, q$ and brain region $k = 1, 2$. The first constraint forces the corresponding
$\Pi_{f,(t,s)}^{kl}$ to zero and thus imposes a banded structure for $\Pi_f^{kl}$, and the second assigns the same sparsity
constraint $\lambda_f$ on the off-diagonal elements of $\Pi_f^{12}$. Finally, to make calibration of tuning parameters
computationally feasible, we set the bandwidth for the latent precisions and the noise precisions
within each region to a single value $h_{\mathrm{auto}}$, we set the bandwidth for the latent precisions across regions
to a value $h_{\mathrm{cross}}$, and we set the sparsity parameters to a value $\lambda_{\mathrm{cross}}$, i.e.,

$$h_f^{kk} = h_\epsilon^k = h_{\mathrm{auto}}, \ h_f^{12} = h_{\mathrm{cross}} \text{ and } \lambda_f = \lambda_{\mathrm{cross}},$$

for each factor $f = 1, \ldots, q$ and region $k = 1, 2$. The bandwidths are chosen using domain
knowledge and preliminary data analyses. We determine the remaining parameters $\lambda_{\mathrm{cross}}$ and $q$ by
5-fold cross-validation (CV).

Solving Eq. (9) requires $S^{-1}$. Because it is not available analytically and a numerical approximation
is computationally prohibitive, we solve Eq. (9) using an EM algorithm (Dempster et al., 1977). Let
$\theta^{(r)}$ be the parameter estimate at the $r$-th iteration. We consider the data $\{X^1[n], X^2[n]\}_{n=1,\ldots,N}$
to be incomplete observations of $\{X^1[n], Z^1[n], X^2[n], Z^2[n]\}_{n=1,\ldots,N}$. In the E-step, we estimate
the conditional mean and covariance matrix of each $\{Z^1[n], Z^2[n]\}$ with respect to $\{X^1[n], X^2[n]\}$
and $\theta^{(r)}$. Given these sufficient statistics, the problem of computing the MPLE decomposes into two
separate minimizations of

1. the negative log-likelihood of $\Sigma_f$, w.r.t. the latent factor model (Eq. (2)) and

2. the negative log-likelihood of $\Phi^1_{\mathcal{S}}$, $\Phi^2_{\mathcal{S}}$, $\Phi^1_{\mathcal{T}}$, $\Phi^2_{\mathcal{T}}$, $\beta^1$, $\beta^2$, $\mu^1$, $\mu^2$ w.r.t. the observation model (Eqs. (1) and (4)).

With the noise correlation and latent factor correlation disentangled, the M-step reduces to easy sub-problems. For example, the minimization with respect to $\Sigma_f$ is a graphical Lasso problem (Friedman et al., 2007) and the minimization with respect to $\Phi^k_{\mathcal{S}}$ and $\Phi^k_{\mathcal{T}}$ is a maximum likelihood estimation of a matrix-variate distribution (Dawid, 1981). We thus obtain an affordable M-step, and alternating E and M-steps produces a solution to the MPLE problem.

We derive the full formulations in Appendix A. Its computational cost is inexpensive: a single iteration of E and M-steps on our cluster server (with 11 Intel(R) Xeon(R) CPU 2.90GHz processors) took in average less than 45 seconds, applied to the experimental data in Section 3.2. A single fit on the same data took 42 iterations for around 30 minutes until P and $\{\beta^1, \beta^2\}$ converged under threshold 1e-3 and 1e-5, respectively. The code is provided at `https://github.com/HeejongBong/ldfa`.

# 3 Results

One major novelty of our method is its accounting for auto-correlated noise in neural time series to better estimate cross-region associations in CCA type analysis. This is illustrated in Section 3.1 based on simulated data. In Section 3.2, we apply LDFA-H to experimental data to examine the lead-lag relationships across two brain areas and the spatial distribution of factor loadings.

## 3.1 LDFA-H retrieves cross-correlations even when noise auto-correlations dominate

We simulated $N = 1000$ i.i.d. neural time series $X^k$ of duration $T = 50$ from Eq. (1) for brain regions $k = 1, 2$. The latent time series $Z^k$ were generated from Eq. (2) with $q = 1$ pair of factors and correlation matrix $P_1$ depicted in Fig. 2a. The noise $\epsilon^k$ was taken to be the $N = 1000$ trials of the experimental data analyzed in Section 3.2, first permuted to remove cross-region correlations, then contaminated with white noise to modulate the strength of noise correlation relative to cross-region correlations. The resulting temporal noise correlation matrices, found by averaging correlations over all pairs of simulated time series, are shown in Fig. 2b, for four levels of white noise contamination. The magnitudes of cross-region correlation and within-region noise auto-correlation are quantified by the determinant of each matrix, known as the generalized variance (Wilks, 1932); their logarithms are provided atop the panels in Fig. 2a and Fig. 2b. Generalized variance ranges from 0 (identical signals) to 1 (independent signals). Thus, larger negative values indicate stronger within-region noise correlation (see B). Other simulation details are in B.

We note that the simulation does not satisfy some of the model assumptions in Section 2. The noise vectors $\epsilon^k$ are not matrix-variate distributed as in Eqs. (4) and (5) and the derived $\Gamma^k_{\mathcal{T}}$ does not satisfy a banded structure as in Eq. (9). Also, the latent partial auto-correlations (Fig. 2) are not banded as assumed in Eq. (9).

We applied LDFA-H with $q = 1$ factor, $h_{\text{cross}} = 10$, $h_{\text{auto}}$ equal to the maximum order of the auto-correlations in the 2000 observed simulated time series, and $\lambda_{\text{cross}}$ determined by 5-fold CV. Fig. 3 shows LDFA-H cross-precision matrix estimates corresponding to the four level of noise correlation shown in Fig. 2b. They closely match the true $\Pi^{12}_1$ shown in the right most panel of Fig. 2a.

We also applied five other methods to estimate cross-region connections in the simulated data. They include the popular averaged pairwise correlation (APC); correlation of averaged signals (CAS); and CCA (Hotelling, 1936), applied to the $NT$ observed pairs of multivariate random vectors $\{X^1_{:,t}, X^2_{:,t}\}_{n,t \in [N] \times [T]}$ to estimate the cross-correlation matrix between the canonical variables; as well as DKCCA (Rodu et al., 2018) and LaDynS (Bong et al. (2020)). The first four methods do not explicitly provide cross-precision matrix estimates, so we display their cross-correlation matrix estimates in Fig. 4, along with LDFA-H cross-correlation estimates in the last row. It is clear that only LDFA-H successfully recovered the true cross-correlations shown in the second panel of Fig. 2a, at all auto-correlated noise levels.

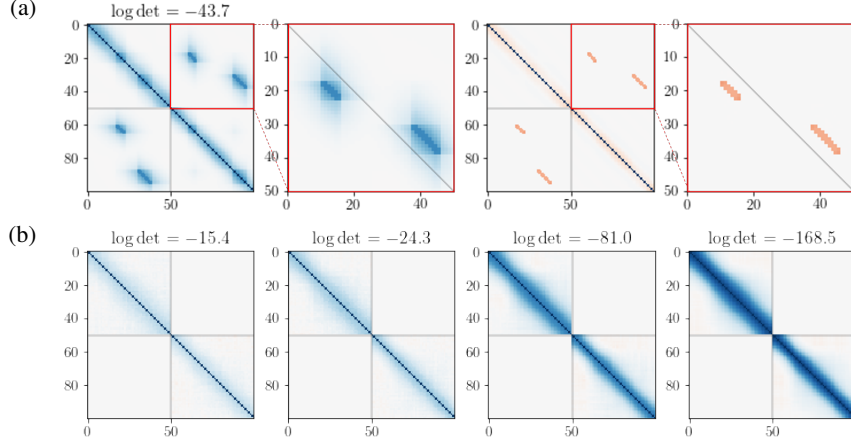

(a)

(b)

Figure 2: **Simulation settings. (a)** *(Left to right panels) True correlation matrix* $P_1$ *for latent factors* $Z^1_{1,:}$ *and* $Z^2_{1,:}$ *from model in Eq. (2); close-up of the cross-correlation matrix; corresponding precision matrix* $\Pi_1 = P_1^{-1}$; *and close-up of cross-precision matrix* $\Pi^{12}_1$ *(Eq. (3)). Matrix axes represent the duration,* $T = 50$ *ms, of the time series. Factors* $Z^1$ *and* $Z^2$ *are associated in two epochs:* $Z^2$ *precedes* $Z^1$ *by 7ms from* $t = 13$ *to 19ms, and* $Z^1$ *precedes* $Z^2$ *by 7ms from* $t = 33$ *to 42ms.* **(b)** *Noise auto-correlation matrices (Eq. (5)) for pairs of simulated time series at four strength levels.* $\log \det$ *in (a) and (b) measure correlation strengths.*

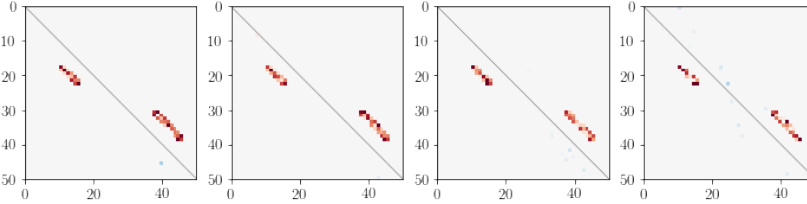

Figure 3: **Simulation results: LDFA-H cross-precision matrix estimates.** *Estimates of* $\Pi^{12}_1$, *shown in the right-most panel of Fig. 2a, using LDFA-H, for the four noise auto-correlation strengths shown in Fig. 2b. LDFA-H identified the true cross-area connections at all noise strengths.*

### 3.2 Experimental Data Analysis from Memory-Guided Saccade Task

We now report the analysis of LFP data in areas PFC and V4 of a monkey during a saccade task, provided by Khanna et al. (2020). One trial of the experiment consisted of four stages: (i) fixation: the animal fixated at the center of the screen; (ii) cue: a cue appeared on the screen randomly at one of eight locations; (iii) delay: the animal had to remember the cue location while maintaining eye fixation; (iv) choice: the monkey made a saccade to the remembered cue location. We focused our analysis on the 500 ms delay period, when the animal both processed cue information and prepared a saccade. LFP data were recorded for $N = 1000$ trials by two 96-electrode Utah arrays implanted in PFC and V4, $\beta$ band-passed filtered, down-sampled from 1 kHz to 100 Hz.

We applied LDFA-H using $h_{\text{auto}} = h_{\text{cross}} = 10$, corresponding to 100 ms (at 100 Hz); the LFP $\beta$-power envelopes have frequencies between $12.5Hz$ to $30Hz$, and $h_{\text{auto}} = 10$ enables the slowest filtered signal to complete one full oscillation period. The other tuning parameters were determined by 5-fold CV over $\lambda_{\text{cross}} \in \{0.0002, 0.002, 0.02, 0.2\}$ and $q \in \{5, 10, 15, 20, 25, 30\}$, yielding optimal values $\lambda_{\text{cross}} = 0.02$ and $q = 10$. We also regularize the diagonal elements, due to the otherwise excessively smooth $\beta$-power envelopes (see our code or Bong et al. (2020) for details). The fitted factors were ranked based on the Frobenius norms of their covariance matrices $\|\Sigma_f\|^2_F$; norms are plotted versus $f$ in decreasing order in Fig. C.1, and $\log_{10} \|\Sigma_f\|^2_F$ of the top three factors are provided above each panel in Fig. 5a. The estimated cross-precison matrices between two brain regions corresponding to the top three factors are shown in Fig. 5a. Note that a positive entry in the precision matrix represents negative association between two regions. We also summarized, for

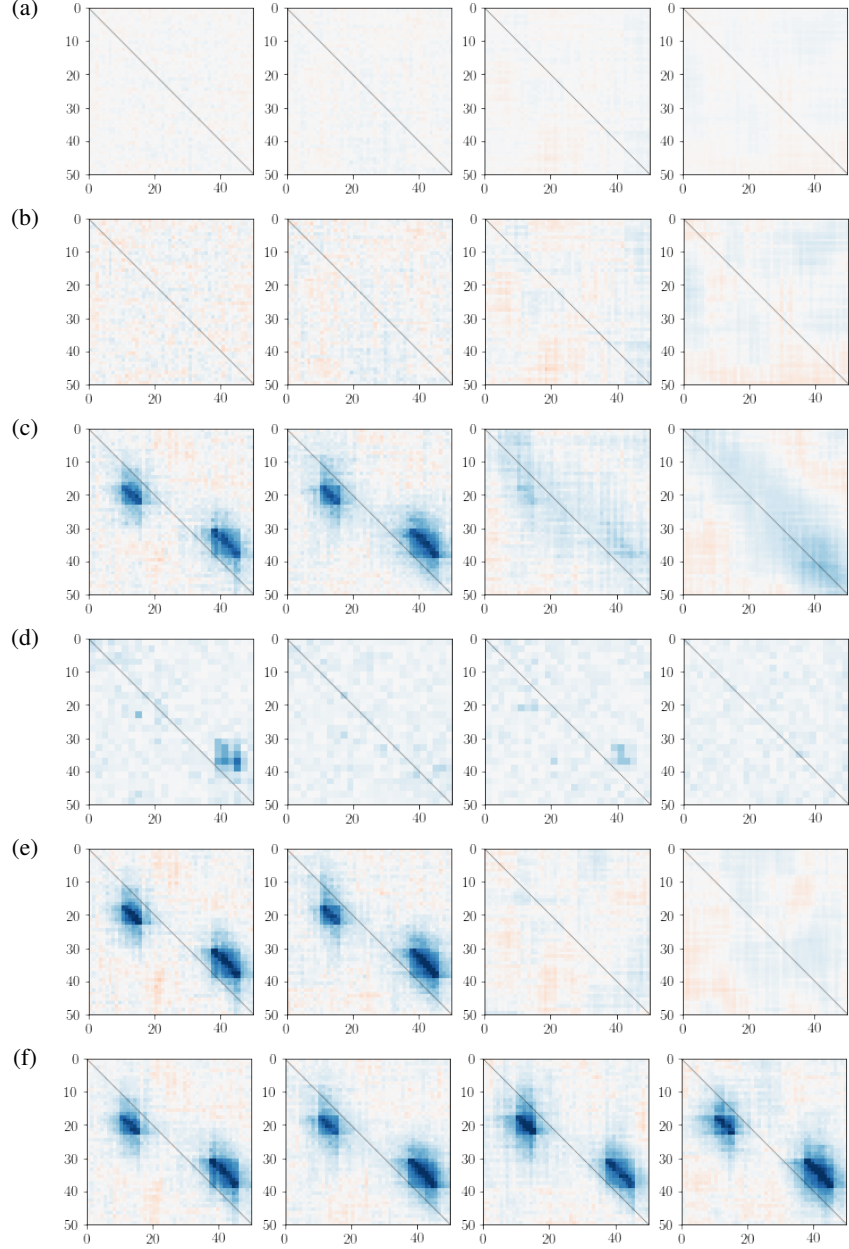

Figure 4: **Simulation results: cross-correlation matrix estimates.** *Estimates of $\Sigma_1^{12}$ under four noise correlation levels using* **(a)** *averaged pairwise correlation (APC),* **(b)** *correlation of averaged signal (CAS),* **(c)** *canonical correlation analysis (CCA, Hotelling (1936)),* **(d)** *dynamic kernel CCA (DKCCA, Rodu et al. (2018)),* **(e)** *LaDynS (Bong et al. (2020)), and* **(f)** *LDFA-H. Only LDFA-H successfully recovered the true cross-correlation at all noise auto-correlation strengths.*

each factor $f$, the temporal information flow at time $t$ from V4 to PFC and to V4 from PFC with $I_{f,PFC \to V4}(t) = \sum_{t'>t} \left| \widehat{\Pi}_{f,(t,t')}^{12} \right|$ and $I_{f,V4 \to PFC}(t) = -\sum_{t'<t} \left| \widehat{\Pi}_{f,(t,t')}^{12} \right|$, respectively, where $\widehat{\Pi}_f$ is the inverse correlation matrix estimate in Eq. (9). Fig. 5d displays smoothed $I_{f,PFC \to V4}(t)$ and $I_{f,V4 \to PFC}(t)$ as functions of $t$ for the top three factors. Lead-lag relationships between V4 and PFC change dynamically over time, and the information flow tends to peak either early in the delay period, when the animal must remember the cue, or later, when it must make a saccade decision. The dominant first factor captures a flow from V4 to PFC centered around 200 milliseconds into the

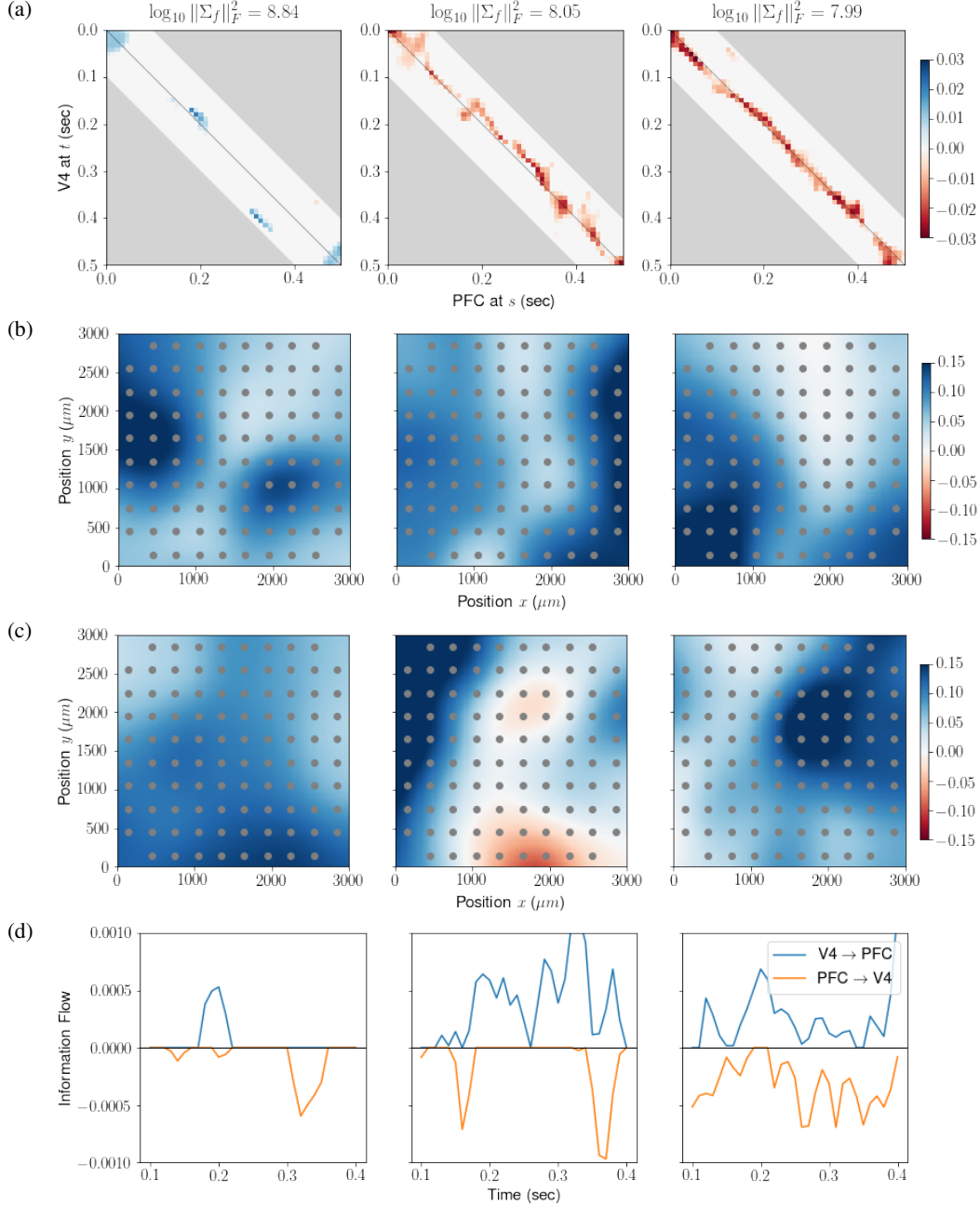

Figure 5: **Experimental data results for the top 3 factors.** *(a) Cross-precision matrices. Blue represents positive precision matrix entries, corresponding to negative association. Factors have different connectivity patterns over the experimental trials.* $\log_{10} \|\Sigma_f\|_F^2$, *written atop the panels, measures the strength of each factor. The first factor is more than 6 times larger than the second and third, and displays activity in V4 leading PFC centered around 200 milliseconds and activity in PFC leading V4 centered around 320 milliseconds post cue disappearance. This is also shown in panel (d). (b,c) Factor loadings, smoothed and color coded, plotted on the electrode coordinates ($\mu m$). Here, positivity is arbitrary, due to identifiability. Panels (b) and (c) display loadings for the V4 and PFC arrays, respectively. The first factor has activity in V4 centered in two distinct subregions of the array, while activity in PFC is more broadly distributed. (d) Dynamic information flow in the directions $V4 \to PFC$ (blue) and $PFC \to V4$ (orange).*

task and a flow from PFC to V4 centered around 320 milliseconds. Factor loadings (subsampled over space) for the 96 V4 and PFC electrodes are shown in Fig. 5b and Fig. 5c, respectively, for the top three factors (first three columns of the estimate of $\beta^k$ in Eq. (9), with area $k = 1$ being V4 and $k = 2$ being PFC), arranged spatially according to electrode positions on the Utah array. The factors have different spatial modes over the physical space of the Utah array. Confirmation of these patterns would require additional data and analyses.

## 4   Conclusion

To identify dynamic interactions across brain regions we have developed LDFA-H, a nonstationary, multi-group extension of GPFA that allows for within-group spatio-temporal dependence among high-dimensional neural recordings. We applied the method to data during a memory task and found interesting, intuitive results. Although we treated the two-group case, and applied it to interactions across two brain regions, several groups can be handled with straightforward modifications. The approach could, in principle, be applied to many different types of time series, but it has some special features: first, like all methods based on sparsity, it assumes a small number of large effects are of primary interest; second, it uses repetitions, here, repeated trials, to identify time-varying dependence; third, because the within-group spatio-temporal structure is not of interest, the method can remain useful even with some modest within-group model misspecification.

Several restrictive assumptions of LDFA-H, as defined, were helpful here but could be modified for other applications. One is the Kronecker-product form of the noise process. In our simulation study, using a realistic scenario, we showed that LDFA-H can be effective even when the Kronecker-product assumption is violated, but in other cases it may be problematic. In some problems, space and/or time can be decomposed into windows within which the assumption is more reasonable (see Leng and Tang, 2012; Zhou, 2014). Another potentially bothersome assumption is independence between latent factors. It would be possible to include covariance matrix parameters between the factors, but then the model will get computationally prohibitive even with a moderate factor size. State-space models (Buesing et al., 2014; Linderman et al., 2019; Yang et al., 2016) have potential but, to be comparable to LDFA-H, they would have to accommodate nonstationary lead-lag behavior. Computationally efficient methods for identifying time-varying relationships is a vital goal in the analysis of neural data from multiple brain regions.

We applied LDFA-H to LFP data. In contrast, GPFA has been applied mainly to neural spike count data, and it is of course possible to apply LDFA-H to spike counts, as well. However, we have been struck by the strong attenuation of effects due to Poisson-like noise, as discussed in Vinci et al. (2018) and references therein. A version of LDFA-H built for Poisson-like counts, or for point processes, could be the subject of additional research. It may also be advantageous to model spatial dependence explicitly, perhaps based on physical distance between electrodes, analogously to what was done in Vinci et al. (2018), and there may be, in addition, important simplifications available in the temporal structure. It would also be helpful to have additional statistical inference procedures for assessing effects. In the future, we hope to pursue these possible directions, and refine the application of this promising approach to the analysis of high-dimensional neural data.

## Broader Impact

While progress in understanding the brain is improving life through research, especially in mental health and addiction, in no case is any brain disorder well understood mechanistically. Faced with the reality that each promising discovery inevitably reveals new subtleties, one reasonable goal is to be able to change behavior in desirable ways by modifying specific brain circuits and, in animals, technologies exist for circuit disruptions that are precise in both space and time. However, to determine the best location and time for such disruptions to occur, with minimal off-target effects, will require far greater knowledge of circuits than currently exists: we need good characterizations of interactions among brain regions, including their timing relative to behavior. The over-arching aim of our research is to provide methods for describing the flow of information, based on evolving neural activity, among multiple regions of the brain during behavioral tasks. Such methods can lead to major advances in experimental design and, ultimately, to far better treatments than currently exist.

## Acknowledgments and Disclosure of Funding

Bong, Liu, Ventura, and Kass are supported in part by NIMH grant R01 MH064537. Smith is supported by NIH (R01EY022928, R01MH118929, R01EB026953, P30EY008098) and NSF (NCS 1734901) grants. Ren is supported in part by NSF grant DMS 1812030.

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
