[Supplementary Material]

# A EM-algorithm to fit LDFA-H (Section 2)

**Initialization**  Let $\widehat{\theta}^{(0)} = \{\widehat{\Sigma}_1^{(0)}, \ldots, \widehat{\Sigma}_q^{(0)}, \widehat{\Phi}_{\mathcal{S}}^{1,(0)}, \widehat{\Phi}_{\mathcal{S}}^{2,(0)}, \widehat{\Phi}_{\mathcal{T}}^{1,(0)}, \widehat{\Phi}_{\mathcal{T}}^{2,(0)}, \widehat{\beta}^{1,(0)}, \widehat{\beta}^{2,(0)}, \widehat{\mu}^{1,(0)}, \widehat{\mu}^{2,(0)}\}$
be the initial parameter value. Since the MPLE objective function for LDFA-H given in Eq. (9) is
not guaranteed convex, an EM-algorithm may find a local minimum according to a choice of the
initial value. Hence a good initialization is crucial to a successful estimation. Here we suggest an
initialization by a canonical correlation analysis (CCA).

Let $\{X^1[n], X^2[n]\}_{n=1,\ldots,N}$ be $N$ simultaneously recorded pairs of neural time series. We can view
them as $NT$ recorded pairs of multivariate random vectors $\{X^1_{:,t}[n], X^2_{:,t}[n]\}_{(n,t)\in[N]\times[T]}$. We obtain
$\widehat{\beta}_1^{1,(0)}$ and $\widehat{\beta}_1^{2,(0)}$ by CCA as follows:

$$\widehat{\beta}_1^{1,(0)}, \widehat{\beta}_1^{2,(0)} = \underset{\beta_1^1 \in \mathbb{R}^{p_1}, \beta_1^2 \in \mathbb{R}^{p_2}}{\operatorname{argmax}} \frac{\beta_1^{1\top} S^{12} \beta_1^2}{\sqrt{\beta_1^{1\top} S^{11} \beta_1^1} \sqrt{\beta_1^{2\top} S^{22} \beta_1^2}} \tag{A.1}$$

where

$$
\begin{aligned}
S^{11} &= \frac{1}{NT} \sum_{n,t} (X^1_{:,t}[n] - \frac{1}{NT} \sum_{n,t} X^1_{:,t}[n])(X^1_{:,t}[n] - \frac{1}{NT} \sum_{n,t} X^1_{:,t}[n])^\top \\
S^{22} &= \frac{1}{NT} \sum_{n,t} (X^2_{:,t}[n] - \frac{1}{NT} \sum_{n,t} X^2_{:,t}[n])(X^2_{:,t}[n] - \frac{1}{NT} \sum_{n,t} X^2_{:,t}[n])^\top \\
S^{12} &= \frac{1}{NT} \sum_{n,t} (X^1_{:,t}[n] - \frac{1}{NT} \sum_{n,t} X^1_{:,t}[n])(X^2_{:,t}[n] - \frac{1}{NT} \sum_{n,t} X^2_{:,t}[n])^\top.
\end{aligned}
\tag{A.2}
$$

According to the equivalence between CCA and probablistic CCA shown by A. Anonymous, it gives
an estimate of the first latent factors

$$\widehat{Z}_{1,:}^{k,(0)}[n] = \widehat{\beta}_1^{k,(0)} X^k[n] \tag{A.3}$$

for $n = 1, \ldots, N$ and $k = 1, 2$. The initial second latent factors $\widehat{Z}_2^{k,(0)}$ and the corresponding
factor loading $\widehat{\beta}_2^{k,(0)}$ is similarly set by the second pair of canonical variables, and so on. Then we
assign the empirical covariance matrix of $\{\widehat{Z}_f^{1,(0)}[n], \widehat{Z}_f^{2,(0)}[n]\}_{n\in[N]}$ to the initial latent covariance
matrix $\widehat{\Sigma}_f^{(0)}$ for $f = 1, \ldots, q$ and the matrix-variate normal estimate (Zhou, 2014) on $\{\widehat{\epsilon}^{k,(0)}[n] :=
X^k[n] - \widehat{\beta}^{k,(0)} \widehat{Z}^{k,(0)}[n]\}_{n\in[N]}$ to $\widehat{\Phi}_{\mathcal{T}}^{k,(0)}$ and $\widehat{\Phi}_{\mathcal{S}}^{k,(0)}$ for $k = 1, 2$. Along $\widehat{\mu}^{k,(0)} := \frac{1}{N} \sum_{n=1}^N X^k[n]$,
the above parameters comprises the initial parameter set $\widehat{\theta}^{(0)}$.

However, we cannot run an E-step on the above parameter set because $\widehat{\Phi}^{k,(0)}$ is not invertible. We
instead pick one of its unidentifiable parameter sets $\widehat{\theta}^{(0),\{\alpha^1,\alpha^2\}}$, defined in Eq. (8), with all $\widehat{\Phi}^{k,(0)}$'s
and $\widehat{\Sigma}_f^{(0)}$'s invertible. Specifically, we take

$$\alpha_f^k = \frac{1}{2} \lambda_{\min} \left( \Sigma_f^{1/2} \begin{bmatrix} \Phi_{\mathcal{T}}^1 & 0 \\ 0 & \Phi_{\mathcal{T}}^2 \end{bmatrix}^{-1} \Sigma_f^{1/2} \right) \tag{A.4}$$

for $f = 1, \ldots, q$ and $k = 1, 2$ where $\lambda_{\min}(A)$ is the smallest eigenvalue of symmetric matrix $A$.
Henceforth, we notate $\widehat{\theta}^{(0),\{\alpha^1,\alpha^2\}}$ by $\widehat{\theta}^{(0)}$. For $t = 1, 2, \ldots$, we iterate the following E-step and
M-step until convergence.

Another promising initialization is by finding time $(t, s)$ on which the canonical correlation between
$X^1_{:,t}$ and $X^2_{:,s}$ maximizes. i.e., we initialize $\widehat{\beta}_1^{1,(0)}$ and $\widehat{\beta}_1^{2,(0)}$ by

$$\widehat{\beta}_1^{1,(0)}, \widehat{\beta}_1^{2,(0)} = \underset{\beta_1^1 \in \mathbb{R}^{p_1}, \beta_1^2 \in \mathbb{R}^{p_2}}{\operatorname{argmax}} \frac{\beta_1^{1\top} S_{(t,s)}^{12} \beta_1^2}{\sqrt{\beta_1^{1\top} S_{(t,t)}^{11} \beta_1^1} \sqrt{\beta_1^{2\top} S_{(s,s)}^{22} \beta_1^2}} \quad \text{such that } |t - s| < h_{\text{cross}}. \tag{A.5}$$

where

$$S^{11}_{(t,t)} = \frac{1}{N} \sum_{n,t} (X^1_{:,t}[n] - \frac{1}{N} \sum_n X^1_{:,t}[n])(X^1_{:,t}[n] - \frac{1}{N} \sum_n X^1_{:,t}[n])^\top$$

$$S^{22}_{(s,s)} = \frac{1}{N} \sum_{n,s} (X^2_{:,s}[n] - \frac{1}{N} \sum_n X^2_{:,t}[n])(X^2_{:,s}[n] - \frac{1}{N} \sum_n X^2_{:,s}[n])^\top \qquad \text{(A.6)}$$

$$S^{12}_{(t,s)} = \frac{1}{N} \sum_{n,t} (X^1_{:,t}[n] - \frac{1}{N} \sum_n X^1_{:,t}[n])(X^2_{:,s}[n] - \frac{1}{N} \sum_n X^2_{:,s}[n])^\top.$$

for $(t,s) \in [T] \times [T]$. Then the other parameters are initialized as above. We can even take an ensemble approach in which we fit LDFA-H on different initialized values and pick the estimate with the minimum cost function (Eq. (9)).

Now, for $r = 1, 2, \ldots$, we alternate an E-step and an M-step until the target parameter $\Pi_f$ convergences.

**E-step** Given $\widehat{\theta} := \widehat{\theta}^{(r-1)}$ from the previous iteration, the conditional distribution of latent factors $Z^1[n]$ and $Z^2[n]$ with respect to observed data $X^1[n]$ and $X^2[n]$ on trial $n = 1, \ldots, N$ follows

$$\left( Z^1_{1,:}[n]; Z^2_{1,:}[n]; \ldots; Z^2_{q,:}[n] \right) \mid X^1[n], X^2[n] \sim \text{MVN} \left( m^{(r)}_{\vec{Z}|X}[n], V^{(r)}_{\vec{Z}|X} \right), \qquad \text{(A.7)}$$

where

$$V^{(r)}_{\vec{Z}|X} = \begin{pmatrix} V^{(r)}_{Z_1,Z_1|X} & \cdots & V^{(r)}_{Z_1,Z_q|X} \\ \vdots & \ddots & \vdots \\ V^{(r)}_{Z_q,Z_1|X} & \cdots & V^{(r)}_{Z_q,Z_q|X} \end{pmatrix} = \begin{pmatrix} W^{(r)}_{Z_1,Z_1|X} & \cdots & W^{(r)}_{Z_1,Z_q|X} \\ \vdots & \ddots & \vdots \\ W^{(r)}_{Z_q,Z_1|X} & \cdots & W^{(r)}_{Z_q,Z_q|X} \end{pmatrix}^{-1} \qquad \text{(A.8)}$$

and

$$\begin{aligned} m^{(r)}_{\vec{Z}|X}[n] &= \left( m^{(r)}_{Z^1_1|X}; m^{(r)}_{Z^1_2|X}; \ldots; m^{(r)}_{Z^2_q|X} \right) \\ &= V^{(r)}_{\vec{Z}|X} \left( \widehat{\beta}^{1\top}_1 \widehat{\Gamma}^1_{\mathcal{S}} X^1[n] \widehat{\Gamma}^1_{\mathcal{T}} ; \ \widehat{\beta}^{2\top}_1 \widehat{\Gamma}^2_{\mathcal{S}} X^2[n] \widehat{\Gamma}^2_{\mathcal{T}} ; \ \ldots ; \ \widehat{\beta}^{2\top}_q \widehat{\Gamma}^2_{\mathcal{S}} X^2[n] \widehat{\Gamma}^2_{\mathcal{T}} \right) \end{aligned} \qquad \text{(A.9)}$$

given

$$W^{(r)}_{Z_f,Z_g|X} = \begin{pmatrix} \left( \widehat{\beta}^{1\top}_f \widehat{\Gamma}^1_{\mathcal{S}} \widehat{\beta}^1_g \right) \widehat{\Gamma}^1_{\mathcal{T}} & 0 \\ 0 & \left( \widehat{\beta}^{2\top}_f \widehat{\Gamma}^2_{\mathcal{S}} \widehat{\beta}^2_g \right) \widehat{\Gamma}^2_{\mathcal{T}} \end{pmatrix} + \mathbb{I}_{\{f=g\}} \widehat{\Omega}_f, \ \ \mathbb{I}_{\{f=g\}} = \begin{cases} 1, & f = g \\ 0, & \text{o.w.} \end{cases} \qquad \text{(A.10)}$$

for $f, g = 1, \ldots, q$.

**M-step** We find $\widehat{\theta}^{(r)}$ which maximize the conditional expectation of the penalized likelihood under the same constraints in Eq. (9), i.e.

$$\begin{aligned} \widehat{\theta}^{(r)} = \underset{}{\arg\min} \ &\frac{1}{N} \sum_{n=1}^N \mathbb{E}_{Z[n]|X[n],\widehat{\theta}^{(r-1)}} \left[ \log p(X^1[n], X^2[n], Z^1[n], Z^2[n]; \widehat{\theta}^{(r-1)}) \right] \\ &+ \sum_{f=1}^q \sum_{k,l=1}^2 \left\| \Lambda^{kl}_f \odot \Pi^{kl}_f \right\|_1 \ \text{ s.t. } \ \widehat{\Gamma}^k_{\mathcal{T}} \text{ is } (2h^k_\epsilon + 1)\text{-diagonal} \end{aligned} \qquad \text{(A.11)}$$

where $p$ is the probability density function of our model in Eqs. (1), (4) and (5) and the expectation $\mathbb{E}_{Z[n]|X[n],\widehat{\theta}^{(r-1)}}$ follows the conditional distribution in Eq. (A.7). Taking a block coordinate descent approach, we solve the optimization problem by alternating M1 - M4.

M1: With respect to latent precision matrices $\Omega_f$, Eq. (A.11) reduces to a graphical Lasso problem,

$$\widehat{\Omega}^{(r)}_f = \underset{\Omega_f}{\arg\min} \left\{ -\log \det(\Omega_f) + \text{tr} \left( \Omega_f \left( V^{(r)}_{Z_f|X} + \widehat{\mathbb{E}}[m^{(r)}_{Z_f|X} m^{(r)\top}_{Z_f|X}] \right) \right) + \sum_{k,l=1}^2 \left\| \Lambda^{kl}_f \odot \Pi^{kl}_f \right\|_1 \right\} \qquad \text{(A.12)}$$

for each $f = 1, \ldots, q$ where $\widehat{\mathbb{E}}[m_{Z_f|X}^{(r)} m_{Z_f|X}^{(r)\top}] = \frac{1}{N} \sum_{n=1}^{N} m_{Z_f|X}^{(r)}[n] \, m_{Z_f|X}^{(r)\top}[n]$. The graphical Lasso problem is solved by the P-GLASSO algorithm by Mazumder et al. (2010).

M2: With respect to $\Gamma^k$, Eq. (A.11) reduces to an estimation of matrix-variate normal model (Zhou, 2014). The estimation problem can be formulated as

$$\widehat{\Gamma}_{\mathcal{S}}^{k(r)} = \frac{1}{T} \left( \widehat{\mathbb{E}}\left[ m_{\epsilon^k|X}^{(r)} m_{\epsilon^k|X}^{(r)\top} \right] + \sum_{f,g=1}^{q} \text{tr}(V_{Z_f^k, Z_g^k|X}^{(r)}) \beta_f^k \beta_g^{k\top} \right) \tag{A.13}$$

and

$$\widehat{\Gamma}_{\mathcal{T}}^{k(r)} = \underset{\Gamma_{\mathcal{T}}^k}{\arg\min} \left\{ \begin{array}{l} - \log\det(\Gamma_{\mathcal{T}}^k) \\ + \dfrac{1}{p_k} \text{tr}\Big( \Gamma_{\mathcal{T}}^k \Big( \sum_{f,g=1}^{q} (\beta_f^{k\top} \Gamma_{\mathcal{S}}^k \beta_g^k) \, V_{Z_f^k, Z_g^k|X}^{(r)} + \widehat{\mathbb{E}}\left[ m_{\epsilon^k|X}^{(r)\top} \Gamma_{\mathcal{S}}^k m_{\epsilon^k|X}^{(r)} \right] \Big) \Big) \end{array} \right\} \tag{A.14}$$

$$\text{s.t. } \widehat{\Gamma}_{\mathcal{T}}^k \text{ is } (2h_\epsilon^k + 1)\text{-diagonal}$$

for each $k = 1, 2$ where $m_{\epsilon^k|X}^{(r)} = X^k - \beta^k m_{Z^k|X}^{(r)} - \mu^k$ and $\widehat{\mathbb{E}}[A]$ is the empirical mean of a random matrix $A$. The estimation of $\Gamma_{\mathcal{T}}^k$ under the bandedness constraint is tractable with modified Cholesky factor decomposition approach with bandwidth $h_\epsilon^k$ using the procedure by Bickel and Levina (2008).

M3: With respect to $\beta^k$, Eq. (A.11) reduces to a quadratic program

$$\widehat{\beta}^{k(r)} = \arg\max_{\beta^k} \left\{ \begin{array}{l} \sum_{t,s} \Gamma_{\mathcal{T},(t,s)}^k \text{tr}\left( \beta^{k\top} \Gamma_{\mathcal{S}}^k \beta_k \, (V_{Z_{:,t}^k, Z_{:,s}^k|X}^{(r)} + \widehat{\text{Cov}}[m_{Z_{:,t}^k|X}^{(r)}, m_{Z_{:,s}^k|X}^{(r)}]) \right) \\ - 2 \sum_{t,s} \Gamma_{\mathcal{T},(t,s)}^k \text{tr}\left( \Gamma_{\mathcal{S}}^k \beta^k \widehat{\text{Cov}}[X_{:,t}^k, m_{Z_{:,s}^k|X}^{(r)}] \right) \end{array} \right\} \tag{A.15}$$

where $\Gamma_{\mathcal{T},(t,s)}^k$ is the $(t,s)$ entry in $\Gamma_{\mathcal{T}}^k$ and $\widehat{\text{Cov}}(A, B)$ is the empirical covariance matrix between random vectors $A$ and $B$. The analytic form of the solution is given by

$$\beta^k = \left( \sum_{t,s} \Gamma_{\mathcal{T},(t,s)}^k (V_{Z_{:,t}^k, Z_{:,s}^k|X}^{(r)} + \widehat{\text{Cov}}[m_{Z_{:,t}^k|X}^{(r)}, m_{Z_{:,s}^k|X}^{(r)}]) \right)^{-1} \left( \sum_{t,s} \Gamma_{\mathcal{T},(t,s)}^k \widehat{\text{Cov}}[m_{Z_{:,s}^k|X}^{(r)}, X_{:,t}^k] \right) \tag{A.16}$$

M4: With respect to $\mu^k$, it is straight-forward that Eq. (A.11) yields

$$\widehat{\mu}^{k(r)} = \widehat{\mathbb{E}}\left[ X^k - \sum_{f=1}^{q} \beta_f^k m_{Z_f^k|X}^{(r)\top} \right].$$

## B  Simulation details (Section 3)

We simulated realistic data with known cross-region connectivity as follows. Simulating $q = 1$ pair of latent time-series $Z^k$ from Equation (2), we introduced an exact ground-truth for the inverse cross-correlation matrix $\Pi_1^{12}$ by setting:

$$\Pi_1 = \begin{bmatrix} (P_{1,0}^{11})^{-1} & 0 \\ 0 & (P_{1,0}^{22})^{-1} \end{bmatrix} + \begin{bmatrix} D^1 & \Pi_1^{12} \\ \Pi_1^{12\top} & D^2 \end{bmatrix} \tag{B.1}$$

where $D^1$ and $D^2$ are diagonal matrices with elements $D_{(t,t)}^1 = \sum_s \Pi_{1,(t,s)}^{12}$ and $D_{(s,s)}^2 = \sum_t \Pi_{1,(t,s)}^{12}$, which ensures that the matrix on the right hand side is positive definite. The matrix on the left hand side contains the auto-precision matrices of the two latent time series, with elements simulated from the squared exponential function:

$$P_{1,0}^{kk} = \left[ \exp\left( -c^k (t-s)^2 \right) \right]_{t,s} + \lambda I_T, \tag{B.2}$$

with $c^1 = 0.105$ and $c^2 = 0.142$, chosen to match the observed LFPs auto-correlations in the experimental dataset (Section 3.2). We added the regularizer $\lambda I_T$, $\lambda = 1$, to render $P^{kk}$ invertible.

Figure C.1: *Squared Frobenius norms of covariance matrix estimates, $\widehat{\Sigma}_f$, for all factors $f = 1, \ldots, 10$. Notice that the amplitudes of the top four factors dominate the others.*

We designed the true inverse cross-correlation matrix $\Pi^{12}$ to induce lead-lag relationship between $Z^1$ and $Z^2$ in two epochs as depicted in the right-most panel of Fig. 2a. Specifically, the elements of $\Pi^{12}$ were set:

$$\Pi^{12}_{(t,s)} = \begin{cases} -r, & \text{where } Z^1_{1,t} \text{ and } Z^2_{1,s} \text{ partially correlate,} \\ 0, & \text{elsewhere,} \end{cases} \quad \text{(B.3)}$$

where the association intensity $r = 0.6$ was chosen to match our cross-correlation estimate in the experimental data (Section 3.2). Finally, we rescaled $P_1 = \Pi_1^{-1}$ to have diagonal elements equal to one. The corresponding factor loading vector $\beta_1^k$ was randomly generated from standard multivariate normal distribution and then scaled to have $\|\beta_1^k\|_2 = 1$.

We generated the noise $\epsilon^k$ from the $N = 1000$ trials of the experimental data analyzed in Section 3.2. First, we permuted the trials in one region to remove cross-region correlations. Let $\{Y^1[n], Y^2[n]\}_{n=1,\ldots,N}$ be the permuted dataset. Then we contaminated the dataset with white noise to modulate the strength of noise correlation relative to cross-region correlations. i.e.

$$\epsilon^k_{:,t} = Y^k_{:,t} - \mu^k_{:,t} + \eta^k_{:,t}, \quad \eta^k_{:,t} \overset{\text{indep}}{\sim} \text{MVN}\left(0, \lambda_\epsilon \widehat{\text{Cov}}[Y^k_{:,t}]\right), \quad \text{and} \quad \mu^k_{:,t} = \widehat{\mathbb{E}}[Y^k_{:,t}] \quad \text{(B.4)}$$

where $\widehat{\mathbb{E}}[Y^k_{:,t}]$ and $\widehat{\text{Cov}}[Y^k_{:,t}]$ wer the empirical mean and covariance matrix of $Y^k_{:,t}$, respectively, for $k = 1, 2, t = 1, \ldots, T$. The noise auto-correlation level was modulated by $\lambda_\epsilon \in \{2.78, 1.78, 0.44, 0.11\}$. We also obtained $\Sigma_1$ by scaling $P_1$ so that $\Sigma^{kk}_{1,(t,s)} = \beta_1^{k\top} S_t^k \beta_1^k$. Putting all the pieces together, we generated observed time series by Eq. (1).

## C    Experimental data analysis details (Section 3.2)

The strength of each factor, which is characterized by $\Sigma_f$, is shown in Fig. C.1.

We also examined an alternative definition of information flow, using non-stationary regresssion in the spirit of Granger causality. For the latent factor $f$ in V4 at time $t$, we use partial $R^2$, effectively comparing the full regression model using the full history of latent variables in both area,

$$Z^1_{f,t} \sim Z^1_{f,1:t-1} + Z^2_{f,1:t-1}$$

with the reduced model using history of latent variables in V4 only,

$$Z^1_{f,t} \sim Z^1_{f,1:t-1}.$$

The partial $R^2$ for $Z^1_{f,t}$ on $Z^2_{f,1:t-1}$ given $Z^1_{f,1:t-1}$ summarizes the contribution of PFC history to V4, after taking account of the autocorrelation in V4, and thus can be viewed as information flow from V4 to PFC at time $t$. Dynamic information flow from V4 to PFC is defined similarly. The results shown in Fig. C.2 are consistent with those in Fig. 5d.

Figure C.2: **Information flow by partial $R^2$ for the top three factors.** *In this figure, we characterize dynamic information flow in terms of partial $R^2$. We show dynamic information flow from $V4 \rightarrow PFC$ (blue) and $PFC \rightarrow V4$ (orange). The results in the first panel are consistent with those in the first panel of Fig. 5d.*