[Reviews · NeurIPS 2020]

Review 1

Summary and Contributions: The paper introduces Latent Dynamic Factor Analysis of High-dimensional time series (LDFA-H), a generalization of Gaussian process factor analysis for studying the interactions of two brain regions. The authors demonstrate the performance of the method on synthetic and real data recorded from PFC and V4.

Strengths: The authors propose an innovative model and inference procedure to generalize GPFA and CCA by allowing the observation noise to have spatiotemporal dependencies and by dropping the stationarity assumption on the latent processes. The computational heavy lifting is done by expectation maximization. To the best of my knowledge, the method is novel. The paper presents the proposed mathematical framework in detail and evaluates it on synthetic data as well as real data. The authors compare the performance of their proposed method to that of a reasonable range of other methods: APC, CAS, CCA, DKCCA and another method by the same anonymous authors. The performance of LDFA-H is impressive, in particular given that the model used to generate synthetic data violates assumptions of LDFA-H. Moreover, results on the real data indicate particular trial times without explicitly providing these times in the training process.

Weaknesses: It is disappointing that the method does not seem to work on neural spike count data as discussed in the Conclusion section. This means that the method is no alternative to GPFA on the most common data that GPFA is used on. The authors state this clearly, so this is no criticism on the presentation of the material. For the noise random vector epsilon, the authors assume that the noise covariance can be expressed as a Kronecker product of purely spatial and purely temporal matrices. While the authors emphasize that they see epsilon as a nuisance variable, the product assumption means that spatio-temporal interactions must be represented by the latent random vector Z. One implication of this assumption for the auto-correlation is briefly mentioned but implications for data analysis are neither explored nor discussed.

Correctness: The description of the model and inference procedures is rigorous. I could not find any technical mistakes, neither in the main paper nor in the Supplemental. The analysis of the synthetic and real data appears to be rigorous as well. Except for the bandwidths, the authors use cross validation for parameter selection and report cross-precision / cross-correlation matrix estimates as well as factor loadings.

Clarity: The paper is very well written and structured. The method is well motivated and the description of the model and inference procedures is clear. Results both for the synthetic data and for the real data are presented well with easily understandable figures. Temporal information flows are estimated by summing over particular precision matrix elements. No references are given for this procedure and limitations and assumptions are not discussed. The authors briefly discuss an alternative method based on R^2 in the Supplemental. Nevertheless, additional details on background and assumptions of the temporal information flow estimation would be helpful in the main paper.

Relation to Prior Work: First and foremost, LDFA-H is related to Gaussian process factor analysis and probabilistic CCA in that it is a generalization of these methods. For the fitting procedures, the authors also encountered a graphical Lasso problem and maximum likelihood estimation of a matrix-variate distribution. The solution to these problems followed prior work (Friedman et al., 2007; Dawid, 1981). The authors also refer to an unpublished article by the same anonymous authors. Reproducibility of the work is outstanding. The authors included a link to an anonymous GitHub repository which contains well documented and organized Python code for LDFA-H as well as Jupyter notebooks and experimental data to reproduce the results. I did not actually run the code but skimmed over it briefly. The paper and Supplemental have the mathematical details of the model and inference procedures including a thorough description of the adapted EM algorithm. Simulations and analysis of experimental data are described in detail as well.

Reproducibility: Yes

Additional Feedback: Edit: It is good to hear that LDFA-H can be applied to spike count data. Clarifying this in the Discussion will improve the paper. Regarding the Kronecker-product assumption and resulting within area interactions, even though the authors claim that the method does not force these interactions, they did not provide enough details in their response, so I feel I cannot raise the score.


Review 2

Summary and Contributions: The authors present a latent factor analysis model to capture the dynamics of variability across neural populations in different brain regions. They derive an EM algorithm to fir the model parameters to large-scale neural recordings.

Strengths: This work builds clearly onto previous methodology to apply more broadly to modeling neural populations across multiple regions. The model definition (including the discussion on identifiability) were sound. The simulations comparing inference to previous methods in Fig 3 were quite compelling (especially the comparison to dkCCA). I believe this methodology to be of broad interest to the NeurIPS community and for those analyzing large-scale neural recordings.

Weaknesses: It’s unclear how much more computationally intensive this method is compared to the other approaches considered. UPDATE: The authors' response showed that this model can be estimated quickly.

Correctness: The claims presented are correct and the methods are appropriate for the target problem. However, the information flow metrics on line 188 were a little unclear to me. More intuition on what’s being computed would be nice. I’m also confused how the formula for I_{f,out}(t), being a sum over positive terms for t’ > t, isn’t a strictly decreasing function (fig 4b), but I could be missing something really simple here. UPDATE: The author's response helped clarify the definition of this metric.

Clarity: The paper is well-written and the methods and data analysis were clear.

Relation to Prior Work: The authors clearly outline how this builds upon previous methods including latent factor analysis and CCA.

Reproducibility: Yes

Additional Feedback: For the sake of completeness, are these spatial maps of the PFC that could be added to fig 4a? Additionally, are there within-area autocorrelations that could be compared to the between area couplings in 4b?


Review 3

Summary and Contributions: This paper introduces a new constrained multivariate normal model for simultaneously recorded neural data (arguably mis-)named Latent Dynamic Factor Analysis of High-dimensional time series. The primary goal of the method is to identify the cross-dependence between two multivariate time-series (here, recordings from two brain areas), allowing for temporal dependence both within and between the two series. The temporal dependence is assumed to be limited in maximum delay (in the domain of precision or partial correlation). Covariance within each area is assumed to decompose into a (Kronecker) product over space and time; covariance between areas is mediated by a reduced number of mutually independent covariant latent time series. The model is fit by EM, assuming a sparse prior on the allowed cross-precision elements. The sensitivity of the method is demonstrated in a simulation, and results are shown for experimental data.

Strengths: This is a useful extension to the suite of multivariate normal methods employed with neural data. Inter-area interactions are an issue of considerable interest, but the potential for delays in influence between areas, and temporal autocorrelation of independent variability within each area, poses a challenge. This model, like many others, proposes a factorisation of the joint covariance of the recordings. The specific factorisation proposed here (combining elements of methods like CCA and GPFA) seems well-suited to answer the question posed.

Weaknesses: Normality is a restrictive assumption, although shared by many other methods (and arguably more reasonable when applied to LFP). I was very confused by the name. The latent factors are only "dynamic" in that temporal correlation is modelled, a property shared with GPFA and others. There is no explicit dynamical system invoked (see point below). I also failed to understand the case for the "H" -- why is this method any better suited to high-dimensional data than others? The scaling with recorded dimension seems no different to standard GPFA, for example. Perhaps the largest concern is the assumption that covariance between areas can be captured entirely by separate latents that are correlated in time and across areas but uncorrelated with each other. A full dynamical model would include potential correlation between latent dimensions -- this is a key difference between the GPFA of Yu et al. (2009) and multivariate state-space models.

Correctness: The model, constraints and prior seem reasonable, though open to criticism as all simple models are. The fitting approach appears to correctly implement likelihood maximisation by EM.

Clarity: I found the paper clear and easy to understand.

Relation to Prior Work: Yes

Reproducibility: Yes

Additional Feedback: Response to author feedback: On non-stationarity: I'm not sure that stationarity was a defining feature of GPFA, but indeed it seems to have been applied that way in most cases. However I agree that having non-stationary interactions between areas explicitly encoded in the cross-precision terms is indeed valuable. On independent latents: I'm not surprised that the model can unmix an instantaneous mixture of two latents -- such linear combinations live within the original model class. However a joint state-space model on Z could create more complex latent interactions, which would be reflected in richer structure in the measured data.


Review 4

Summary and Contributions: This manuscript proposes a new modeling framework to capture interdepencies between groups of high-dimensional data and applies the methodology in the context of Local Field Potentials in a repeated trial formulation. In my opinion, the primary contribution is the modeling framework and the EM algorithm to efficiently approximate the conclusions. The scientific results (Figure 4) are an additional contribution that reveal interesting patterns in the V4 response to stimuli.

Strengths: This topic is relevant to the NeurIPS community since it is a situation that should increase in importance as dense multi-electrode arrays are being implanted in the brain, LFP analysis is increasing in importance, and the majority of work on implanted electrodes focuses on spikes. The modeling and algorithm contributions seem sound and a reasonable approach. The scientific results are intriguing, seem reasonable, but are challenging to compare to ground truth.

Weaknesses: My primary issue with the work presented is the lack of comparisons and the limited discussion of related work, so the presented work does not appear in context. Notably, there are already works that attempt to relate groups of observations, and there are many works that attempt to estimate information flow between LFPs (e.g. [1], [2], [3], [4], etc.), and there exist group factor analysis approaches that can account for interactions (e.g., [5]). To my knowledge, the proposed approach is different than what is in the literature, but needs to be heavily revised to discuss what the differences between the proposed approach and the existing literature are. It's interesting that the authors do not adjust their model depending on the variable stimuli. This modeling choice should be explained and motivated. The complete lack of comparisons and an ablation study hinders this manuscript. Are all the assumptions necessary? What do they do to the results if they remove them? Is this model improving what we can conclude compared to existing approaches? The utility of the scientific analysis is not motivated. What is the hypothesis that this method is answering? What novel information do we glean scientifically? How will the learned information be used? Details on the anonymous unpublished work should have been included with the supplemental material; as is, it is very difficult to understand the relationship to that work and it is difficult to evaluate the differences. Update after review: without rereading the revised manuscript it is difficult to tell if they will my criticisms will be really addressed or not (rebuttal simply says that they will do it). While I do view the proposed work as novel, I view the relationships to existing work as closer than claimed. [1] Adhikari, Avishek, et al. "Cross-correlation of instantaneous amplitudes of field potential oscillations: a straightforward method to estimate the directionality and lag between brain areas." Journal of neuroscience methods 191.2 (2010): 191-200. [2] Jiang, Haiteng, et al. "Measuring directionality between neuronal oscillations of different frequencies." Neuroimage 118 (2015): 359-367. [3] Hultman, Rainbo, et al. "Brain-wide electrical spatiotemporal dynamics encode depression vulnerability." Cell 173.1 (2018): 166-180. [4] Gallagher, Neil, et al. "Cross-spectral factor analysis." Advances in Neural Information Processing Systems. 2017. [5] Buesing, Lars, et al. "Clustered factor analysis of multineuronal spike data." Advances in Neural Information Processing Systems. 2014.

Correctness: They appear correct.

Clarity: The manuscript is mostly well-written. Revising to improve clarity and the claims would benefit the manuscript; section 2 appears a bit abstract at first, and providing an example of the data here could help make the section grounded and easier to understand. Revising to motivate the scientific question a bit more would improve the manuscript as well.

Relation to Prior Work: No, see "Weaknesses."

Reproducibility: Yes

Additional Feedback: It would benefit the manuscript to discuss how/whether this is extendable to multiple groups.

[Author Response · NeurIPS 2020]

We thank the reviewers for insightful comments and address the major issues they raised.

**Reviewer 1**   **Weaknesses:** (a) LDFA-H can be applied to spike count data in its current form, and we will change
our discussion to clarify; we can also demonstrate its use in a spike count simulation. The misleading statement in
our discussion should have said, more specifically, that we believe a better approach (for future work) would be to add
an additional Poisson layer to the model, for reasons discussed in Vinci et al (2018) and related references. (b) The
Kronecker product assumption does not force within-area interaction into the latent vector. Through simulations we
have investigated spatio-temporal interactions under our Kronecker product assumption, and in our revision we will
show how to handle them in the supplementary material.

**Reviewer 2**   **Weaknesses:** Although we have been unable to find a good way to make direct comparisons with other
methods, the method here is reasonably fast and we will note that a single fit of the real data took less 19 seconds
on a 2.2 GHz single-processor MacBook. **Correctness:** The definition of information flow in our real data section is
intuitive, yet it is also subject to judgement. Because neural time series can be treated as vector auto-regressive, in the
supplementary material we will provide partial $\chi^2$ as another way to characterize information flow. In our revision,
we also plan to add figures showing $\hat{\Pi}_f^{12}$ to further reveal the dynamic connectivity between two areas. Notice that
$\hat{\Pi}_f^{12}$ is banded, and we are calculating $I_{f,out}(t)$ starting from $100ms$ (the bandwidth). The quantity $I_{f,out}(t)$ does
not necessarily increase with $t$ because it involves summation of non-zero entries only within the band. **Feedback:**
We will add figures for PFC in the supplementary material. In addition, we will also add within-area connectivity for
comparison with cross-area connectivity.

**Reviewer 3**   **Weaknesses:** (a) LDFA-H allows latent connectivity to be non-stationary, and thus, unlike GPFA,
its structure can evolve across time; in this sense it is dynamic. It deals explicitly with the high dimensionality of
non-stationary time series within areas by assuming a sparse, banded, Kronecker product structure. We will make this
clearer. (b) The reviewer makes the important point that LDFA-H assumes the correlation across areas occurs in matched
latent variables, so that differing components remain independent across areas, and the reviewer notes this may be
restrictive. A key observation is that even in cases when this assumption is incorrect, the method can identify cross-area
correlation, but, as we have now shown in a simple simulation, the situation is somewhat subtle. We simulated $Z_1'$, $Z_2'$
from $\Sigma_1'$, $\Sigma_2'$ according to our model with one area leading the other area for $Z_1'$ and the reverse lead-lag relationship for
$Z_2'$. When then combined the latents in the form $Z_1 = 0.6 * Z_1' + 0.4 * Z_2'$, $Z_2 = 0.4 * Z_1' + 0.6 * Z_2'$, and re-computed
$\Sigma_1$ and $\Sigma_2$ for each factor. The corresponding $\Pi_f^{12}$ indicates (intuitively) *both* lead-lag relationships. To demonstrate
recovery we generated data from $Z_1$ and $Z_2$ and applied our method. It produced factors that separately pull out the two
lead-lag relationships which, if combined, tell the ground-truth story about lead-lag in $Z$. The reviewer is thus correct
that interpretation requires care, and we will say so in our revision, including the simulation results in the supplementary
material. Furthermore, in future work we plan to generalize the approach to avoid this restrictive assumption, and will
say so.

**Reviewer 4**   **Weaknesses:** (a) We should have included discussion of additional related work, including the articles
the reviewer mentions. In our revision we will explain that none of those methods is directly comparable to ours. For
example, we agree it could be beneficial to apply a spectral approach (though in sliding windows to deal with non-
stationarity), but that is complementary to ours in the sense that it aims for connectivity among oscillatory components:
our view is that pulses of activity often have temporal profiles that are better described in the time domain. (b)
Concerning key assumptions, please see above. (c) In our revision we will articulate the main novel purpose and ability
of our approach: again, it can find connectivity for pulses of activity that are not accurately described by stationary
processes. (d) We will add details about the anonymous work. **Clarity:** We will introduce an example in section 2.
**Feedback:** we will discuss the extension to multiple groups, which is straightforward.

[Meta-Review · NeurIPS 2020]

Reviewers have somewhat variable opinions on this paper. Reviewer 4 feels it needs to be better distinguished from the literature. Ablation analyses that reveal more about what aspects are necessary would also strengthen the paper. Reviewer 3 raises a valid concern about limiting assumptions in the model - "an instantaneous mixture of latents is not the sort of dependence that a state-space model would capture". Reviewers 1 and 2 are more positive on the paper and most of their earlier concerns were addressed in the rebuttal. This paper is close to the boundary for acceptance.